# Machine Learning Based Predictions of Dissolved Oxygen in a Small Coastal Embayment

**Manuel Valera [1,*]**, **Ryan K. Walter [2]**, **Barbara A. Bailey [1]** and **Jose E. Castillo [1]**

1   Computational Science Research Center, San Diego State University, San Diego, CA 92182, USA;
    bbailey@sdsu.edu (B.A.B.); jcastillo@sdsu.edu (J.E.C.)
2   Physics Department, California Polytechnic State University, San Luis Obispo, CA 93407, USA;
    rkwalter@calpoly.edu
*   Correspondence: mvalera-w@sdsu.edu

**Abstract:** Coastal dissolved oxygen (DO) concentrations have a profound impact on nearshore ecosystems and, in recent years, there has been an increased prevalance of low DO hypoxic events that negatively impact nearshore organisms. Even with advanced numerical models, accurate prediction of coastal DO variability is challenging and computationally expensive. Here, we apply machine learning techniques in order to reconstruct and predict nearshore DO concentrations in a small coastal embayment while using a comprehensive set of nearshore and offshore measurements and easily measured input (training) parameters. We show that both random forest regression (RFR) and support vector regression (SVR) models accurately reproduce both the offshore DO and nearshore DO with extremely high accuracy. In general, RFR consistently peformed slightly better than SVR, the latter of which was more difficult to tune and took longer to train. Although each of the nearshore datasets were able to accurately predict DO values using training data from the same site, the model only had moderate success when using training data from one site to predict DO at another site, which was likely due to the the complexities in the underlying dynamics across the sites. We also show that high accuracy can be achieved with relatively little training data, highlighting a potential application for correcting time series with missing DO data due to quality control or sensor issues. This work establishes the ability of machine learning models to accurately reproduce DO concentrations in both offshore and nearshore coastal waters, with important implications for the ability to detect and indirectly measure coastal hypoxic events in near real-time. Future work should explore the ability of machine learning models in order to accurately forecast hypoxic events.

**Keywords:** dissolved oxygen; random forest; support vector machine; machine learning regression; hypoxia

## 1. Introduction

Coastal dissolved oxygen (DO) concentrations have a profound impact on nearshore ecosystems and biogeochemical cycling [1]. When DO concentrations drop below a certain threshold, macroorganisms are stressed and, in some cases, the so-called hypoxic conditions can lead to massive mortality of ecologically and economically important fish and invertebrates [2–4]. Coastal hypoxia has emerged as a major threat in coastal ecosystems, due to the increased frequency and duration of hypoxic events [3]. Furthermore, as climate change further drives ocean-warming deoxygenation of the world's oceans, a better predictive capacity for the onset of hypoxic conditions and dissolved oxygen variability is needed.

Applications of machine leaning techniques in the field of oceanography have substantially increased in recent years, particularly as a supplementary computational tool for model assessment

with varied applications, including sediment transport [5], remote sensing [6–11], local sea-level fluctuations [12,13], equation of state calculations [14], and surface gravity wave modeling [6,15,16]. In many instances, machine learning algorithms have become more efficient and computationally cheaper and faster than traditional hydrodynamic and biogeochemical numerical models [6,14–18]. Moreover, machine learning techniques are able to easily replicate nonlinear phenomena from a sufficiently large dataset with an appropriate number of features and target quantities. When tuning and training a machine learning model in a certain geographic extent with a range of input variables and target variable, they are able to accurately predict the target variable while only using the basic input variables. Thus, machine learning models can serve as a near real-time monitoring and prediction tool for a complicated, and difficult to measure, target quantity while using only basic, and easily measured, input parameters. Machine learning models also have the ability to be further refined and trained with more input data, presumably improving their predictive potential. These capabilities have been demonstrated for predicting a range of biogeochemical parameters in various coastal and ocean environments, including chlorophyll concentrations in coastal lagoons [19,20], hypoxic risk using geomorphological and bathymetric features in the coastal regions of the Baltic Sea [21], and nutrients and carbonate system variables in the Mediterranean Sea [18], among others ([22–25] and the references therein). In many cases, even with the most advanced numerical models, accurately modeling these biogeochemical parameters is challenging and computationally expensive and, therefore, not as potentially useful as data-driven models and machine learning techniques for near real-time estimates. Despite this, the testing and application of machine learning techniques to predict variables in a range of coastal environments, with varied datasets, across different depths, and with simple and widely measured input parameters, is still relatively limited, particularly for nearshore DO (cf. [25]).

The problem of predicting a possible hypoxic event from data can be solved while using two main approaches. The first approach would involve transforming the response variable to labeled categories by empirically estimating the thresholds of low, medium, or high hypoxia risk [19,21]. This approach simplifies the machine learning model to a classification problem, at the cost of introducing uncertainty to the prediction in the form of the empirically decided thresholds for each label. The second approach, which treats the problem as a regression model, is more comprehensive and it is the one that we have taken in this research [20,22]. In this approach, the hypoxic risk, which can be determined when the DO concentration drops below a certain threshold (e.g., [1]), can be completed either by post-processing or automated in real time. By adopting this approach, we focus on obtaining the best accuracy possible for predicting DO values over their full range for development of a real-time tool that can be used in order to assess hypoxic risk with readily measured parameters. Seawater DO content is sometimes predicted by fitting a linear regression model to offshore data using the more readily measured temperature as a predictor variable given the strong covariation between the two in some situations [26]. However, these relationships are often region specific, requiring a new relationship to be derived for each location. Additionally, simple linear (and even nonlinear) relationships have some degree of uncertainty and cannot be used for accurate predictions in different conditions than those originally measured.

Here, we apply a suite of machine learning techniques in order to predict nearshore DO concentrations and, therefore, hypoxic risk, in a small coastal embayment (San Luis Obispo Bay in Central California—see Section 2.1) using a unique set of both nearshore and vertically-resolved offshore data. Predicting hypoxic events in nearshore coastal waters presents an array of challenges, due to complex dynamical processes and the lack of comprehensive observational data (see e.g., [25]). Moreover, in nearshore coastal systems, embayments are a common feature that indent the coastline and contribute to complicated alongshore variability in physical and biogeochemical water properties [27–30]. We restrict our analysis to a set of input parameters that are readily available and easily measured, following recommendations from a recent review that identified the need to simplify the training features space to widely available parameters [5]. We demonstrate the ability of several machine learning models in order to predict nearshore DO variability, show that only a small subset of

training data is needed to obtain high accuracy predictions, and explore the ability of a site-specific model to predict DO content at adjacent sites.

## 2. Materials and Methods

### 2.1. Study Site and Data

San Luis Obispo (SLO) Bay is a small, semi-enclosed coastal embayment that is located along the Central California Coast (Figure 1). SLO Bay features considerable ecological diversity, including giant kelp forests, a local fishing port, and several tourist destinations. Similar to other embayments in eastern boundary current upwelling systems, seasonal coastal upwelling shapes the physical, chemical, and biological environment of these systems [30–33]. Upwelling bays are particularly susceptible to regional extremes in DO, due to the combination of strong upwelling that advects subthermocline waters low in DO into these systems, high biological productivity, and elevated retention times [27,33,34]. In particular, in SLO Bay, several recent hypoxic episodes have resulted in adverse effects, including fish kills. As part of a larger project investigating the physical drivers of nearshore hypoxia in SLO Bay, a suite of nearshore fixed moorings were deployed and periodic offshore hydrographic transects were completed. This larger project was the first comprehensive field program aimed at investigating the DO dynamics and hypoxic events in SLO Bay. These nearshore and offshore datasets are utilized here.

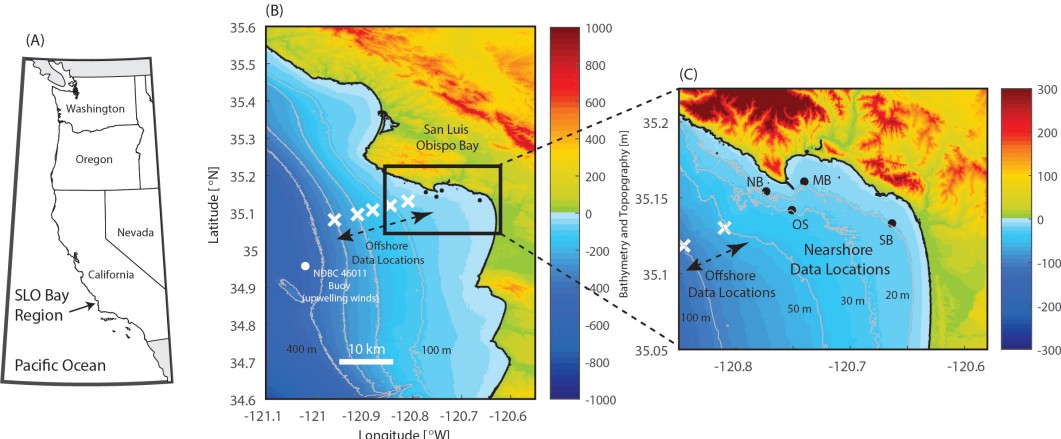

**Figure 1.** (**A**) United States West Coast showing the San Luis Obispo (SLO) Bay region located along the Central California Coast. (**B**) Regional view showing the location of the offshore hydrographic transects (white X), the regional buoy used for the upwelling wind stress, and isobaths from 100 m to 400 m (gray lines). (**C**) Local view of SLO Bay highlighting the four mooring locations (black dots) and isobaths at 20, 30, 50, and 100 m (gray lines).

Four nearshore moorings (OS in approximately 30 m water depth; MB, SB, and NB in approximately 20 m water depth) were deployed in and around SLO Bay during the major upwelling season from late March to late August in 2017 (Figure 1). Each mooring was equipped with a near-bottom (fixed two meters above the bed) Sea-Bird 37 MicroCAT conductivity-temperature-depth (CTD) sensor with a Sea-Bird 63 DO optical optode, sampling every ten minutes. The second component of the field measurements consisted of a series of approximately monthly offshore CTD/DO casts at fixed stations located in depths of 300, 200, 150, 100, and 50 m. These vertical profiles were conducted while using a Sea-Bird 19+ V2 profiling CTD with a Sea-Bird 43 electrochemical profiling DO sensors and bin-averaged every meter. A total of five offshore transects were completed while the nearshore moorings were deployed (April 17, May 22, June 23, July 11, August 8). From these moorings and offshore transects, time, temperature, salinity, depth, and DO measurement data were obtained for use in the machine learning models.

In order to examine the role of wind-driven coastal upwelling, hourly offshore winds were obtained from NDBC buoy 46011 [35] (Figure 1). Equatorward upwelling favorable winds were calculated while using the local coastline orientation [30] and upwelling wind stresses were calculated following [30,36] and utilized in the models.

## 2.2. Selected Models

The first part of our analysis demonstrates the ability of the machine learning techniques to fit a model prediction of DO using only the offshore CTD data at various depths, which is expected to behave in a more linear manner compared with the nearshore data. The second part of this study uses nearshore data in order to extend the machine learning models into a highly nonlinear regime. We first present the model results separately for each individual nearshore mooring site to compare each site. We also explore the possibility of using one nearshore site's DO data to predict the DO contents at the other sites. We also perform a sensitivity analysis of the final model accuracy results in a change in the amount of training data included in the machine learning models to investigate the utility of machine learning models in filling data gaps in the field when a sensor fails to record data (or data are removed due to quality control flags). Last, we present the results for a combined dataset using all of the nearshore site data, while using the CTD data and time as predictors.

Random Forests Regression (RFR) and Support Vector Machines Regression (SVR) are the models selected for our analyses. Random Forests was selected as one of the most prominent machine learning methods being used in ocean sciences [5,7,8,10,12,16,17,19,21,23,24,37,38]. Random Forests is typically favored for its bootstrapping power which enables it to tackle highly nonlinear problems, as well as its decision tree-based methodology that yields a measure of feature importance, a valuable tool in feature selection and dimension-reduction. Support vector machines are selected as an alternative non-tree-based method [5,16,20,22,39–41], which is relatively underrepresented in applications of machine learning to environmental data. We decided to focus on these two techniques, as opposed to other more complex methods, like artificial neural networks, for their straight forward approach and potential wider applicability. The strengths and particularities of each algorithm are discussed in the next section. The sci-kit learn software is used in this work for machine learning algorithms, as well as model scoring and data selection and sampling for training and testing [42].

### 2.2.1. Random Forest Regression

Random forest is a machine learning algorithm that consists of an ensemble of decision trees [43]. Each decision tree is constructed while using recursive partitioning that bases the split points in the tree on each variable, determining the split that increases the homogeneity of the resulting subsets. Final predictions of the random forest are made by averaging the predictions of each decision tree. In the random forest procedure, each tree in the ensemble is grown while using a different bootstrap sample of the original data. Because bootstrapping uses random sampling with replacement, there is approximately one-third of the data that is "out-of-sample" or "out-of-bag" for each tree. This out of sample data acts as an internal validation dataset and it can be used in order to estimate predictions and prediction error, as well as determine the variable importance for each predictor variable. In addition, when growing each tree, a small random subset of the candidate variables available is considered for splitting at each node of the tree. For highly correlated data, the set of randomly selected variables tends to decorrelate the trees and produces more diverse trees. All of these characteristics have helped to popularize random forest as a reliable, wide spectrum prediction algorithm that is used across a range of applications in environmental and biological sciences.

In particular, for this work, we implement random forest regression (RFR), which predicts a continuous quantity as a target, making the algorithm useful in a wider range of applications, but more difficult to tune and obtain accurate results. Table 1 shows the parameters that are used by the RandomForestRegressor scikit-learn function other than the defaults. Both offshore and nearshore cases were implemented with the same parameter configuration, and out-of-bag tests were carried out in order to obtain the optimal parameters for the models. The number of tree used in the ensemble for the RFR is 400 independent trees. The splitting criterion used is mean square error and the random subset of the candidate features used for splitting at each node of the tree is the natural logarithm of the total number of features in each tree. Variable importance is determined while using a node impurity measure (also known as Gini importance). This measure is the decrease in node impurities from splitting on a variable, or the reduction of the impurity that is gained by introducing a split on a specific node, averaged over all trees in the forest.

**Table 1.** Random Forest Regression parameters used other than default.

| RFR | Offshore | Nearshore |
|---|---|---|
| n_estimators (trees) | 400 | 400 |
| criterion | mse | mse |
| max_features | log2 | log2 |

### 2.2.2. Support Vector Regression

Support vector machine methods are part of the class of hyperplane-based methods [39,41]. They are based on the idea of determining optimal separating hyperplanes where the data can be classified. Hyperplanes are decision boundaries and the dimension of the hyperplane depends on the of number features. The algorithm can be extended to a regression problem by considering the data points that are within a small distance of the decision boundary line, and the best fit is the hyperplane that includes the maximumum number of data points. In support vector regression (SVR), the most important parameter is the kernel or decision function used to find the shape of the separating hyperplane. Kernels can take different shapes and complexity, such as linear, polynomial, sigmoid, or the most popular radial-basis function (RBF), and they will define a geometrical area in each hyperplane that belongs to a specific label or value.

For SVR, we can formalize the problem, as follows: given training vectors $x_i$ and a response vector $y$, we want to solve the minimization problem:

$$min(\frac{1}{2}w^T w + C \sum_{i=1}^{n}(\zeta_i + \zeta_i *)) \tag{1}$$

$$subject\ to\ \ y_i - w^T \phi(x_i) - b \le \epsilon + \zeta_i, \tag{2}$$

$$w^T \phi(x_i) - b - y_i \le \epsilon + \zeta_i *, \tag{3}$$

$$\zeta_i \zeta_i * \ge 0,\ i = 1, \ldots, n \tag{4}$$

where $w$ is the solution to the minimization problem $w = (X^T X)^{-1} X^T y$ with $X$ as the feature matrix, $\zeta_i$ and $\zeta_i *$ are the factors that penalize the hyperplane region errors that are either above or below the desired target, $C$ is a parameter that controls the width of the area of the $\epsilon$ tube or separation plane, and $\phi(x_i)$ is the kernel function that maps the data from the input space into the features space, where the problem is solved [17,42].

The function for each kernel used will have specific parameters. In order to obtain the most accurate prediction in SVR, each one of these parameters needs to be tuned by trial and error or various other tuning methods. For this work, we used the scikit-learn SVR algorithm with the nonlinear RBF kernel ($\phi(x) = exp(-\gamma||x - x'||^2)$), where $\gamma$ is set to one over the number of features. Table 2 depicts the rest of the specified parameter values for each model. In this case, the process to find the optimal values is more exhaustive. The SVR complexity grows quadratically with the number of samples and, thus, takes a much longer time to train than RFR. Additionally, each kernel can add further overhead to the calculations. After an exhaustive grid search strategy, we obtained the parameters that are shown in Table 2 for offshore and nearshore cases.

**Table 2.** Support Vector Regression parameters used other than default.

| SVR | Offshore | Nearshore |
|---|---|---|
| Kernel | RBF | RBF |
| $C$ | 1000 | 100 |
| $\gamma$ | 0.20 | 0.20 |
| $\epsilon$ | 0.0031 | 0.019 |

## 3. Results

In order to illustrate the inherent complexity of each dataset, their value distribution and pairwise relationships between variables can be inspected. These pairwise plots are visual representations of a dataset in two-dimensional (2D) sections, which are shown as an array of plots of paired features. Each plot of the array is a pairwise relation of features labeled by row or column number. In matrix notation the *i,j* plot denotes the relationship between quantities of row *i* and column *j*. The main diagonal represents a density histogram of each respective quantity in column *j*. These pairwise relationships highlight the correlation between quantities and, more importantly, can establish linear and nonlinear relationships between features. A linear dataset will show linear relationships on these pairwise plots (or pairplots) and this in return would be regarded as a lower complexity dataset to be trained for and predict on. On the other hand, a nonlinear dataset will show diffuse relationships in the pairplots, being a visual representation of the harder problem to solve when there are no evident relationships between the quantities. In this work, the offshore dataset is closer to linear in nature, while the nearshore dataset is highly nonlinear. The pair-based plots for each of the datasets can be seen in Figure 2a,b with a histogram of value frequency shown in the main diagonal of the plots grid. The time dimension is formatted in days since the 1st January 0000.

We also included a correlation matrix, which highlights the Pearson correlation coefficient for the different variable combinations. Figure 3 shows this matrix. The offshore data display coefficients closer to $-1$ and $+1$ for most of the variables indicating a more negative or positive linear relationship, respectively, except for the upwelling wind stress and time variables, which are much more sparse. The nearshore data show correlation coefficients that are closer to zero for most combinations, indicating the lack of a linear relationship.

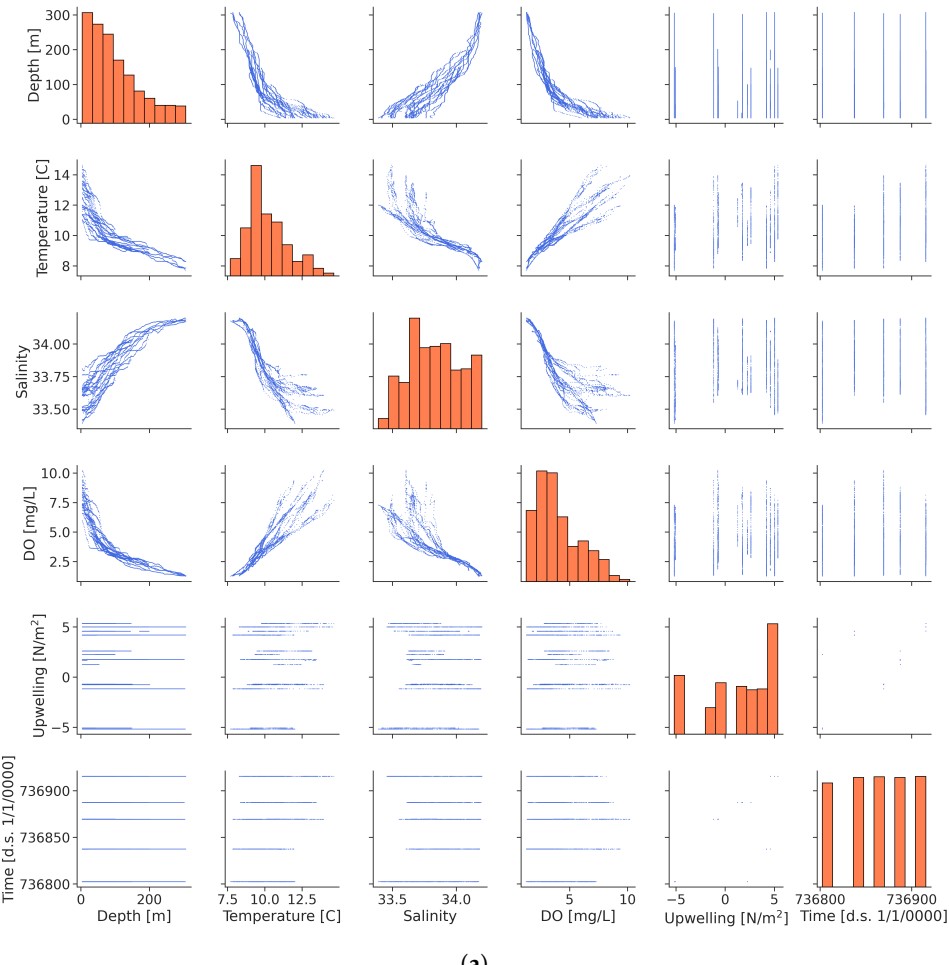

(**a**)

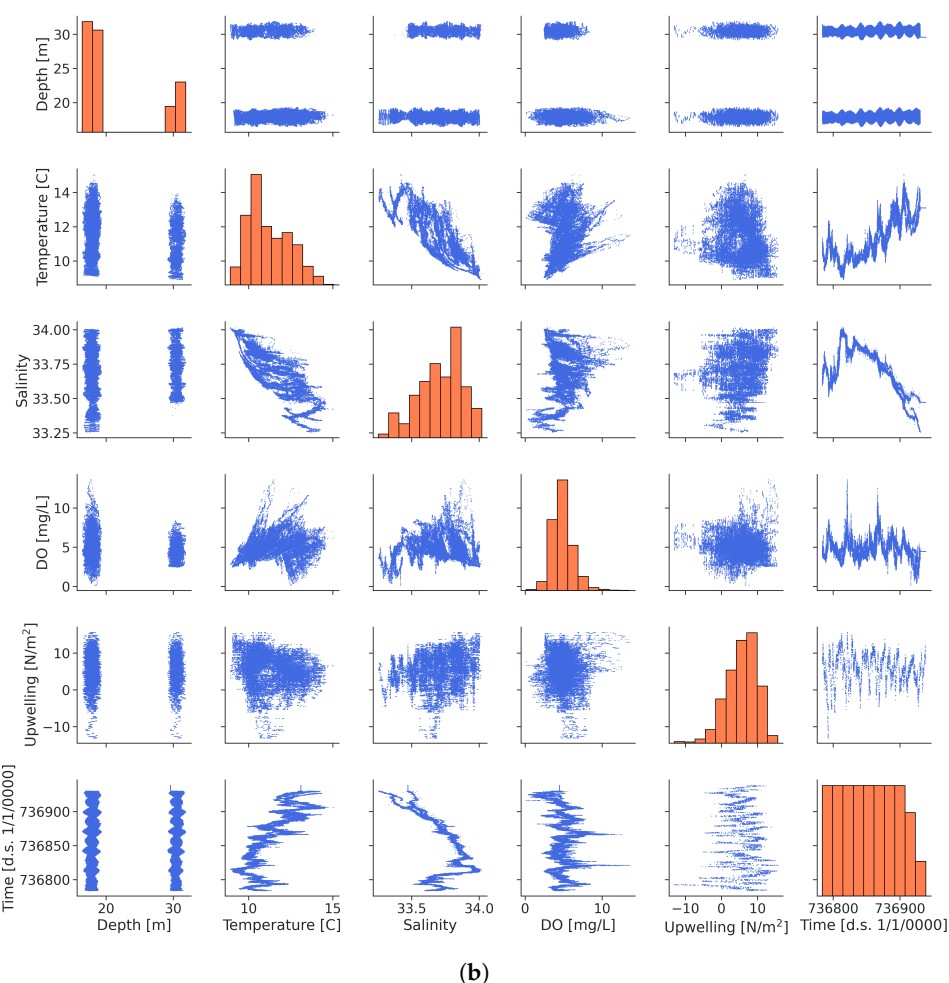

(**b**)

**Figure 2.** Pairwise relationships between all featured variables for the (**a**) offshore data and (**b**) nearshore data. Each plot shows a two-dimensional (2D) scatter plot of the variables labeled at that specific row and column. A histogram of value frequency for each quantity is shown in the main diagonal of the plots grid.

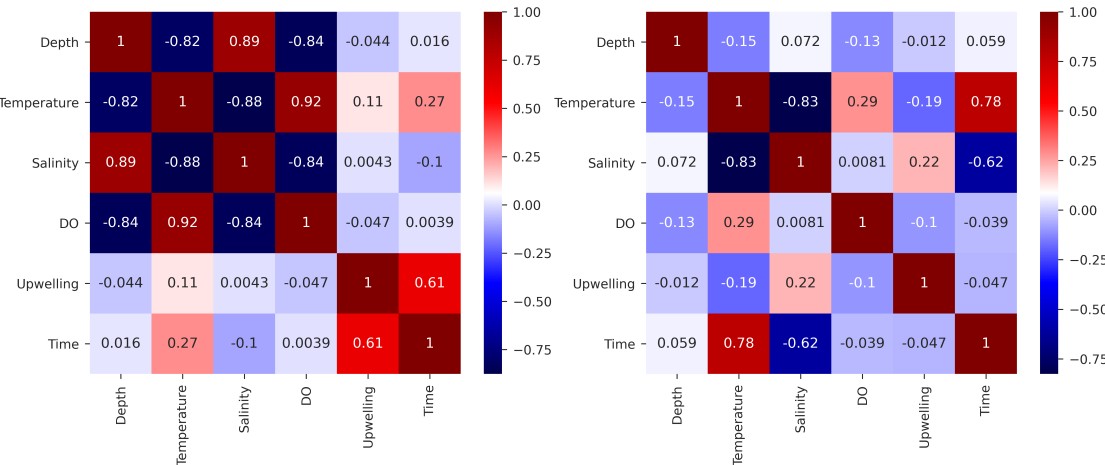

**Figure 3.** Correlation matrices for the offshore dataset (**left**) and nearshore dataset (**right**) for each variable combination. The Pearson correlation coefficient is given by the colorbar and it is annotated inside each grid on the matrix.

### 3.1. Offshore DO Predictions

The RFR and SVR models were trained on the offshore dataset as a proof of concept and a way to generalize and automate the DO estimation tools currently available. These tools use time, temperature, salinity, depth, and upwelling wind stress as tabulated data in order to estimate the DO content at the same time. The training was performed while using 66.67% of the dataset (which included DO), and the prediction or model validation was done using the remaining 33.33% of the data. Training and prediction sets were sampled at random with the `train_test_split` function from scikit-learn.

The results for offshore data are presented in Figures 4 and 5, with the error metrics tabulated in Table 3, including the coefficient of determination ($R^2$), mean absolute error (mae), and mean square error (mse). The predicted data are highly accurate for both RFR and SVR, with $R^2$ values of 0.997 and 0.986, respectively. The RFR showed lower mae and mse relative to SVR. Moreover, the residuals show high accuracy at lower DO values, where the hypoxic risk is greatest, particularly for the RFR. For the RFR, depth was the most important variable while using the node impurity importance, followed by temperature and then salinity. Time and upwelling wind stress were relatively unimportant for these datasets; however, it is important to note that the offshore data were limited to the five discrete partial days of sampling, whereas the nearshore mooring instruments sampled over the entire experiment.

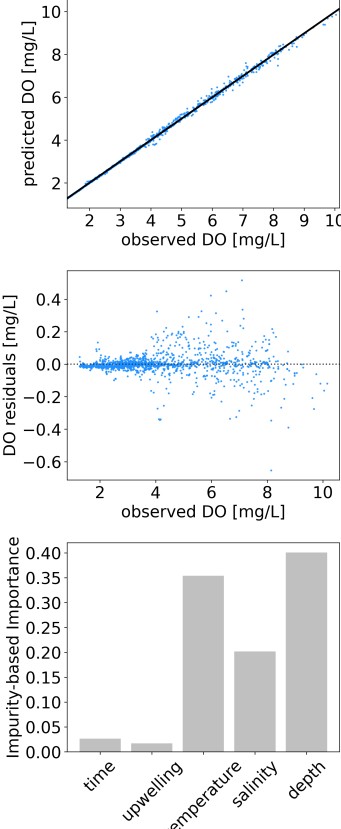

**Figure 4.** Offshore model relational plot for Random Forest Regression (**top**) with $R^2 = 0.997$, along with the residuals plot (**center**) and relative impurity-based importance (**bottom**).

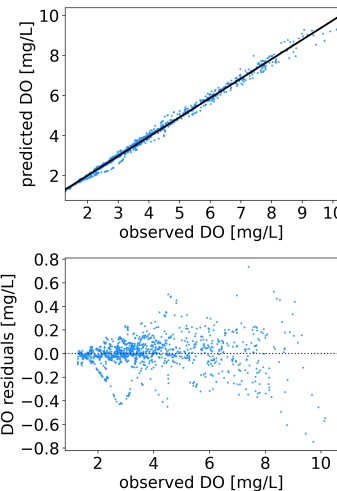

**Figure 5.** Offshore model relational plot for Support Vector Regression (**top**) with $R^2$ = 0.986, along with the residuals plot (**bottom**).

**Table 3.** Offshore oxygen prediction errors results.

|  | RFR | SVR |
|---|---|---|
| $R^2$ | 0.997 | 0.986 |
| mae [mg/L] | 0.044 | 0.089 |
| mse [mg$^2$/L$^2$] | 0.007 | 0.045 |

*3.2. Nearshore DO Predictions*

3.2.1. Nearshore Station-Based Predictions

Prior to using the machine learning models to estimate DO over all nearshore sites, we used the models at each nearshore site individually. We then explored the ability of the models to estimate DO at other sites while using OS and MB as the training datasets. These predictions were all done using RFR due to superior scores and decreased wall time. For self-training at an individual site, 66.67% of the data was used for training and the remaining 33.33% for prediction. For predicting other sites, 99.9% of the data were used from the training site dataset. Table 4 tabulates the error metric results for these different combinations.

**Table 4.** Station-based random forest regression (RFR) error predictions, where the top row shows training the nearshore site used for training and bottom row the nearshore site that is predicted.

| Train: | Self | | | | OS | | | MB | |
|---|---|---|---|---|---|---|---|---|---|
| Predict: | OS | SB | NB | MB | SB | NB | MB | SB | NB |
| $R^2$ | 0.997 | 0.993 | 0.995 | 0.990 | 0.426 | 0.567 | 0.554 | 0.701 | 0.438 |
| mae [mg/L] | 0.027 | 0.071 | 0.040 | 0.080 | 0.857 | 0.493 | 0.676 | 0.600 | 0.632 |
| mse [mg$^2$/L$^2$] | 0.002 | 0.015 | 0.005 | 0.020 | 1.351 | 0.509 | 0.913 | 0.704 | 0.661 |

For the self-training and predictions (i.e., same site), the model at each respective site performed extremely well, with an $R^2$ greater than 0.99 at all sites. The mae ranged from 0.027 to 0.08 mg/L and the mse ranged from 0.002 to 0.02 mg$^2$/L$^2$, with OS performing the best (lowest) and MB performing the worst (highest) for these error metrics. Overall, predictions using training data from the same site performed well.

When using the deeper nearshore site (OS) as the training dataset to predict DO at the other three nearshore sites (SB, NB, MB; see Figure 1), the predictive capabilities were significantly diminished,

with the $R^2$ values ranging between 0.426 and 0.567, mae values between 0.493 and 0.857 mg/L, and mse values between 0.509 and 1.351 mg$^2$/L$^2$. When using the MB nearshore site, which is in the same depth as SB and NB, the predictive results were more mixed with an $R^2$ of 0.438 and 0.701, a mae of 0.600 and 0.634 mg/L, and a mse of 0.704 and 0.661 mg$^2$/L$^2$ at SB and NB, respectively. Contrary to the self-training, these cross-site training/predictions show mixed results and they are more limited in their potential application, depending on the specific error tolerance that is acceptable. This is likely driven by the high non-linearity and site specific dynamics in the different regions of the coastal embayment.

We also explored while using varying percentages of training data in the same site training and prediction RFR models (Figure 6). The training was completed using two distinct feature sets: one set that included the CTD data (temperature, salinity, and depth), time, and upwelling wind stress, and one that only included the CTD data (temperature, salinity, and depth) and upwelling wind stress to explore the impact of removing time as a training variable. In general, these results show that only a small percentage of training data is needed in order to accurately reconstruct DO fields. When including time as an input to the model, an $R^2$ of greater than 0.9 was seen at all sites with as little as 4% of training data. When excluding time, the amount of training data needed to achieve an $R^2$ of greater than 0.9 at all sites increased to 15%, although there was moderate variation across the sites (see Figure 6).

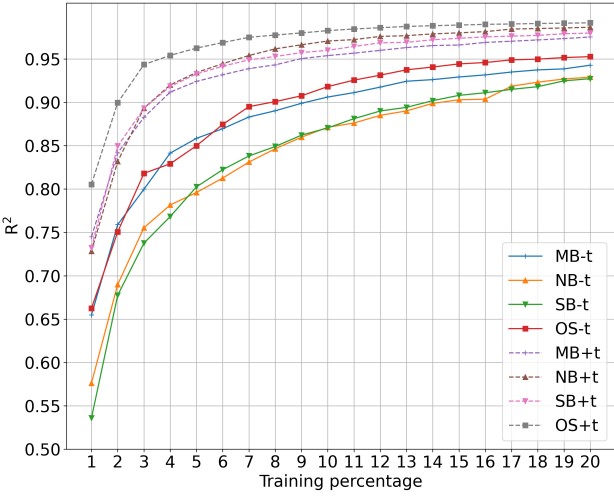

**Figure 6.** Nearshore station-based $R^2$ scores as a function of percentage of available data used to train the model. "+t" series denote the time used as an input in the model.

### 3.2.2. Combined Nearshore Model Results

Finally, the results of aggregating all of the nearshore datasets from all sites together as one training dataset while using 66.67% of the data for training and the remaining 33.33% for validation are shown in Figure 7 for RFR and Figure 8 for SVR, respectively, with the error metrics for both models shown in Table 5. Here, we use the CTD data (temperature, salinity, and depth), time, and upwelling wind stress for the predictions.

For these models, RFR and SVR both performed well and produced accurate predictions. The RFR performed slightly better with an $R^2$ of 0.987 as compared to 0.946 in the SVR model. Likewise, the mae (0.076 mg/L) and mse (0.022 mg$^2$/L$^2$) were better (i.e., lower) in the RFR when compared to the SVR (0.0182 mg/L and 0.091 mg$^2$/L$^2$ for mae and mse, respectively). While the overall model performed well, there were still limited instances when the residuals in the model predictions reached more than 2 mg/L. For the RFR, the relative impurity-based importances showed that time was the most important variable, followed closely by temperature and then salinity. Upwelling wind stress and

depth were not as important, the latter of which is not surprising, given that these are fixed moorings sampling at one nominal depth (i.e., changes in depth are only due to the tidal height changes, which will be consistent across the moorings).

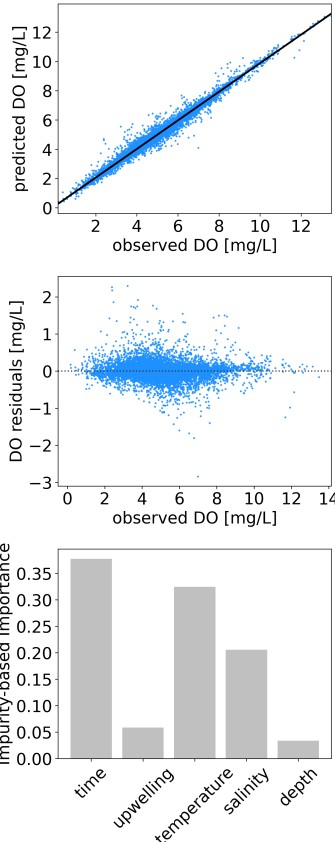

**Figure 7.** Nearshore model relational plot for Random Forest Regression (**top**) with $R^2 = 0.987$, along with the residuals plot (**center**) and relative impurity-based importance (**bottom**).

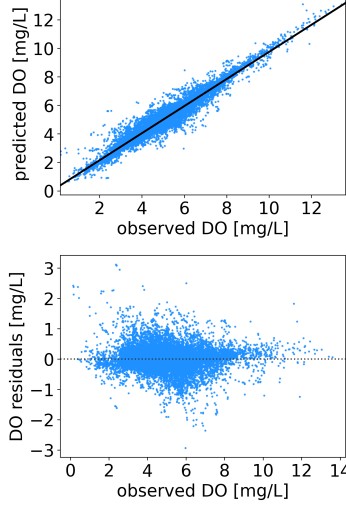

**Figure 8.** Nearshore model relational plot for Support Vector Regression (**top**) with $R^2 = 0.945$, along with the residuals plot (**bottom**).

**Table 5.** Nearshore oxygen prediction error results.

|  | RFR | SVR |
|---|---|---|
| $R^2$ | 0.987 | 0.946 |
| mae [mg/L] | 0.076 | 0.182 |
| mse [mg$^2$/L$^2$] | 0.022 | 0.091 |

## 4. Discussion and Conclusions

We demonstrate that DO predictions in offshore and nearshore coastal waters can be accurately attained while using machine learning regression techniques (RFR and SVR) and are readily available, and easily measured, oceanographic data. For the more linear offshore data set that we tested, both RFR and SVR achieved highly accurate ($R^2 > 0.98$) results with minimal error residuals. This was particularly true for the lower DO values, which allows for the accurate detection of hypoxic events. The methods used here are a good candidate for indirectly determining DO concentrations and hypoxia risk while using readily measured parameters (temperature, salinity, depth, time, and upwelling wind stress), with the appropriate training data.

In the nearshore region, we expected the model accuracy and performance to decrease, given the complex underlying processes and nonlinear nature. When training and predicting DO concentrations with data from one particular site, the machine learning technique (RFR) was extremely accurate, with $R^2$ values that were greater than 0.99 in all cases. However, when using training data from one site to predict another nearshore site, there was only moderate success with $R^2$ values ranging between 0.426 and 0.701, with much higher error residuals. It is well known that coastline orientation and presence of coastal embayments can lead to complex circulation patterns and strong along-shore variability in physical and biogeochemical water properties (see [33] and the references therein). The moderate and highly variable cross-site model success is likely due to the strong spatial gradients in biogeochemical parameters found near coastal embayments and the underlying dynamics at some sites being more similar (e.g., inside the embayment versus outside the embayment). It is possible that the models would substantially improve with additional input variables that are known to affect DO variability (e.g., chlorophyll, stratification, etc.), although further research is needed in this respect. When combining all of the nearshore sites into one dataset, both RFR and SVR were able to very accurately reproduce DO concentrations ($R^2$ values of 0.987 and 0.946, respectively). In general, RFR consistently performed better than SVR, the latter of which was harder to tune and took longer to train.

Additionally, we carried out a series of tests on the minimum required amount of data that are needed for training to achieve varying degrees of accuracy. We found that only minimal amounts of data were needed in the training datasets to achieve high accuracy (e.g., an $R^2$ of greater than 0.9 was seen at all sites with as little as 4% of training data when including time as a training variable). This finding is particularly important, as it indicates that machine learning techniques can be used as a technique for accurately reproducing DO concentrations with relatively little training data. These machine learning techniques could be applied to time series datasets where DO data are missing, due to instrument drift, biofouling, instrument failure, or other quality control issues. The same techniques could also likely be applied to other biogeochemical variables, like pH.

Several previous studies have utilized machine learning techniques for predicting DO variability and hypoxia. Ref. [44] successfully trained neural networks while using a cross-wavelet analysis technique and several decades of yearly-averaged hypoxic volumes in Chesapeake Bay. Ref. [25] used a data-driven model that combined empirical orthogonal functions and neural networks to predict DO in Chesapeake Bay. [45] also used long-term data from Chesapeake Bay and a decision tree machine learning algorithm in order to successfully predict DO variability. Using neural networks and support vector machines, ref. [22] successfully modeled DO while using several years of bimonthly measurements of 11 physical, biological, and chemical input parameters in the Wen-Rui Tan River. All of the aforementioned studies, and others (cf. [25] and the references therein), had success in using a

variety of machine learning techniques in order to predict DO variability or hypoxia risk; however, these studies typically involved years to decades of data and utilized bimonthly to yearly-averaged data as model input. Long-term and comprehensive datasets from many coastal and estuarine systems are not readily available, as noted by [25]. This, in addition to the complex underlying dynamics underpinning DO variability and hypoxia, makes it challenging to accurately simulate and predict DO variability in these environments. This work establishes the ability of straightforward machine learning techniques in order to accurately and indirectly reproduce DO concentrations in both offshore and nearshore coastal waters while using readily available and easily measured parameters. While the approach presented here was specifically for a coastal region in central California, the methodology should be readily applicable to other coastal systems and locations with similar datasets, although further research is needed in this regard. This has implications for the ability to detect and indirectly measure coastal hypoxic events in near real-time, with important implications for management options. Future work will explore the forecasting abilities of machine learning models in order to predict hypoxic events into the future, which will rely on longer sets of training data.

**Author Contributions:** Conceptualization, M.V., R.K.W.; methodology, M.V., R.K.W., B.A.B.; software, M.V.; validation, M.V.; formal analysis, M.V., R.K.W., B.A.B. and J.E.C.; investigation, M.V. and R.K.W.; resources, R.K.W. and J.E.C.; data curation, R.K.W.; writing–original draft preparation, M.V. and R.K.W.; writing–review and editing, M.V., R.K.W., B.A.B. and J.E.C.; visualization, M.V., R.K.W.; supervision, B.A.B. and J.E.C.; project administration, J.E.C.; funding acquisition, R.K.W. and J.E.C. All authors have read and agreed to the published version of the manuscript.

**Funding:** This research was supported by NOAA Grant #NA14OAR4170075, California Sea Grant College Program Project #R/ HCE-07, through NOAA's National Sea Grant College Program, U.S. Department of Commerce. The statements, findings, conclusions and recommendations are those of the authors and do not necessarily reflect the views of California Sea Grant, NOAA, or the U.S. Dept. of Commerce.

**Acknowledgments:** We thank Ian Robbins, Jason Felton, Tom Moylan, Kristin Davis, Grant Waltz, and Wesley Irons for their help in the field. Boating resources were provided by the Cal Poly Center for Coastal Marine Sciences.

**Conflicts of Interest:** The authors declare no conflict of interest.

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
