# Peer review of "Machine Learning Based Predictions of Dissolved Oxygen in a Small Coastal Embayment"

_jmse, doi:10.3390/jmse8121007_

Round 1
Reviewer 1 Report
This paper explores the prediction of dissolved oxygen (DO) in a coastal area of Central California using two machine learning techniques (RFR and SVR). In general, this manuscript is a straightforward study but presents interesting results and I think it would be suitable for publishing. However, I suggest improving the following points:
- In the introduction, add a sentence about where the study is carried out and indicate if any previous studies about OD estimation have been carried out in the area or not. Add the justification about why the prediction of OD in San Luis Obispo Bay is important.
- The authors must provide a justification on the selection of the two machine learning algorithms and why not others were considered.
- The authors should improve the discussion of the results by interpreting them in the perspective of previous studies.
Specific comments:
- The abstract must be reduced. A single paragraph of about 200 words maximum.
- Figure 1, figure 3 and figure 5 have to be placed after being referred to in the text and not before.
- Figure 1: include the map of North America showing the location of the study area and a legend.
- Line 84: regarding nearshore moorings, is OS correct in the text or should it be OB as shown in Figure 1?
- In section 2.1 (Study Site and Data): add a summary table of the characteristics of the data used in the study (range, no. of data collected, correlation coefficient of each variable with OD and any other information that may be relevant).
- Lines 101-117: I find it more appropriate to move this to the introduction section.
- Line 120: Remove the reference to Figure 2a and 2b since these have not yet appeared in the text as they belong to the results section. Figures should be placed in the main text near to the first time they are cited.
- Figure 2: reduce the size of the figures 2a y 2b but without reducing the size of the axis lettering. Give one colour to the diagonal histograms and another to the scatter plots (see eg Figure 3 by Altomare et al. (2020) https://doi.org/10.3390/jmse8080570)
- Figure 3, Figure 4, Figure 6 and Figure 7: Colour the points on the graphs to highlight them against pure lines (line 1:1) and dashed lines (y=0).
- Table 4: Why are the results of MB training not shown in the table to predict OS? If this prediction was not made what was the reason?
- Figure 5: colouring the lines to make them easier to distinguish from each other.
Author Response
Editor 1 - General comments:
This paper explores the prediction of dissolved oxygen (DO) in a coastal area of Central California using two machine learning techniques (RFR and SVR). In general, this manuscript is a straightforward study but presents interesting results and I think it would be suitable for publishing. However, I suggest improving the following points:
- In the introduction, add a sentence about where the study is carried out and indicate if any previous studies about OD estimation have been carried out in the area or not. Add the justification about why the prediction of OD in San Luis Obispo Bay is important.
We thank the reviewer for this suggestion. In the Introduction, we added reference to the study site in San Luis Obispo (SLO) Bay in Central California. Furthermore, we added additional details about SLO Bay in Section 2.1, when describing the study site, including “SLO Bay features considerable ecological diversity, including giant kelp forests, a local fishing port, and several tourist destinations…In particular, in SLO Bay, several recent hypoxic episodes have resulted in adverse effects, including fish kills...This larger project was the first comprehensive field program aimed at investigating DO dynamics and hypoxic events in SLO Bay.”
- The authors must provide a justification on the selection of the two machine learning algorithms and why not others were considered.
An extended justification of the selected models was added to section 2.2
- The authors should improve the discussion of the results by interpreting them in the perspective of previous studies.
A paragraph was added to the discussion section on this.
Specific comments:
- The abstract must be reduced. A single paragraph of about 200 words maximum.
We reduced the abstract significantly from 351 words to 269 words.
- Figure 1, figure 3 and figure 5 have to be placed after being referred to in the text and not before.
Mentioned figures have been placed after being referenced in the text.
- Figure 1: include the map of North America showing the location of the study area and a legend.
The map of the US West Coast showing the location of the study area was added to Figure1.
- Line 84: regarding nearshore moorings, is OS correct in the text or should it be OB as shown in Figure 1?
Thank you for catching this. It should be “OS”, and this was changed in the figure.
- In section 2.1 (Study Site and Data): add a summary table of the characteristics of the data used in the study (range, no. of data collected, correlation coefficient of each variable with OD and any other information that may be relevant).
Added a new figure with the correlation coefficients of each dataset, as well as an explanation of this plot at the start of the results section. The data ranges are shown in the pairwise plots, which have been extended to show them on both axes.
- Lines 101-117: I find it more appropriate to move this to the introduction section.
Moved the stated lines to the introduction.
- Line 120: Remove the reference to Figure 2a and 2b since these have not yet appeared in the text as they belong to the results section. Figures should be placed in the main text near to the first time they are cited.
Removed reference to Figures 2a and 2b here.
- Figure 2: reduce the size of the figures 2a y 2b but without reducing the size of the axis lettering. Give one colour to the diagonal histograms and another to the scatter plots (see eg Figure 3 by Altomare et al. (2020) https://doi.org/10.3390/jmse8080570)
Figures 2a and 2b have been expanded to show the full matrix of pairplots, color has been added to follow more closely the provided reference. The figures size has been diminished and the ticks font size has been increased.
- Figure 3, Figure 4, Figure 6 and Figure 7: Colour the points on the graphs to highlight them against pure lines (line 1:1) and dashed lines (y=0).
The aforementioned figures have had color added as suggested.
- Table 4: Why are the results of MB training not shown in the table to predict OS? If this prediction was not made what was the reason?
OS is in deeper water (~30 m) compared to the shallower MB, SB, and NB (~20 m). Thus, when using MB (a more nonlinear regime compared to OS), as the training dataset, we opted to only include predictions at the other nearshore sites (SB and NB).
- Figure 5: colouring the lines to make them easier to distinguish from each other.
Added color to the plot series.
Reviewer 2 Report
Manuscript ID jmse-1000725
Title Machine learning based predictions of dissolved oxygen in a small coastal embayment
General comment
The manuscript presents the application of two machine-learning techniques, the Random Forest Regression and the Support Vector Regression, to the prediction of the concentration of dissolved oxygen in the nearshore area of a case-study coastal embayment.
The topic is particularly interesting and suitable for the target of this Special Issue. The manuscript is well organized and accurately written. Overall, the reading is fluid, though sometimes details and explanations are needed to get a better understanding of the work. The results are clearly presented and discussed. It is particularly appreciated the open-minded approach adopted by the authors in the investigation of different techniques and solutions to achieve the best results, discussing each time limits and benefits. Overall, it is found that the manuscript is worth of publication once addressed the following minor comments.
Specific Comments.
- Abstract: please, introduce the meaning of the acronyms (RFR, SVR, etc.) before using it.
- Section 3, lines 190-198 and Figures 2a and 2b: this paragraph is quite complicated to follow. The authors are invited to give more information/explanation about the “pairwise relationships” and “pairwise plots” and about the utility of introducing such analyses. Expand the description of Figures 2a and 2b. It is also not clear what the histograms in these Figures represent.
- Lines 206-214 and Figure 3-bottom: specify what is the node impurity measure. Why it is not shown also for SVR (Figure 4)?
- Lines 238-246: the importance of the time-variable seems remarkable for the near-shore case. Can you explain why? Why, on the contrary, time seems to be the least important variable for off-shore dataset, according to Figure 3-bottom?
Author Response
Editor 2 - Specific Comments.
- Abstract: please, introduce the meaning of the acronyms (RFR, SVR, etc.) before using it.
Added acronyms after first enunciation of the methods
- Section 3, lines 190-198 and Figures 2a and 2b: this paragraph is quite complicated to follow. The authors are invited to give more information/explanation about the “pairwise relationships” and “pairwise plots” and about the utility of introducing such analyses. Expand the description of Figures 2a and 2b. It is also not clear what the histograms in these Figures represent.
An explanation of the pairwise plots has been included in the paragraph. The description of Figures 2a and 2b has been expanded accordingly.
- Lines 206-214 and Figure 3-bottom: specify what is the node impurity measure. Why it is not shown also for SVR (Figure 4)?
The explanation for node impurity importance is found before at line 163. The text at line 227 was altered to say “node impurity importance” instead of measure. Importance information is inherent of decision tree methods as is stated in section 2.2.1 of the paper, this is why SVR doesn’t show importances.
- Lines 238-246: the importance of the time-variable seems remarkable for the near-shore case. Can you explain why? Why, on the contrary, time seems to be the least important variable for off-shore dataset, according to Figure 3-bottom?
For the offshore datasets, sampling only occurred on five discrete days. For the nearshore datasets, sampling occurred throughout the entire experiment (~5 months). In Section 3.1, we have the following sentence: “Time and upwelling wind stress were relatively unimportant for these datasets; however, it is important to note that the offshore data were limited to the five discrete partial days of sampling, whereas the nearshore mooring instruments sampled over the entire experiment.”
Reviewer 3 Report
Congratulations to the authors of the manuscript entitled Machine learning-based predictions of dissolved oxygen in a small coastal embayment, The scientific content of the document is impeccable. The Machine learning for DO prediction represents a new frontier of application compared to traditional numerical and biogeochemical models. Machine learning offers more possibilities in terms of flexibility, applicability, and expandability than classical models. The Manuscript in the form in which it has been presented is ready to be published in the JMSE MDPI Journal.
Author Response
Editor 3 - No comments
The authors want to thank the kind words of reviewer number three. These are very encouraging for a young and early career researcher.